# Can the Cans: Determinants of Container Deposit Behavior before and after Introduction of a Container Refund Scheme

**DOI:** 10.3390/bs14020112

**Published:** 2024-02-02

**Authors:** Daniel J. Phipps, Daniel J. Brown, Martin S. Hagger, Kyra Hamilton

**Affiliations:** 1School of Applied Psychology, Griffith University, 176 Messines Ridge Road, Mount Gravatt, QLD 4122, Australia; d.phipps@griffith.edu.au (D.J.P.); daniel.brown@unisq.edu.au (D.J.B.); mhagger@ucmerced.edu (M.S.H.); 2Faculty of Sport and Health Sciences, University of Jyväskylä, 40600 Jyväskylä, Finland; 3School of Psychology and Wellbeing, University of Southern Queensland, 11 Salisbury Road, Ipswich, QLD 4305, Australia; 4Health Sciences Research Institute, University of California, Merced, 5200 North Lake Rd., Merced, CA 95343, USA; 5Department of Psychological Science, University of California, Merced, 5200 North Lake Rd., Merced, CA 95343, USA

**Keywords:** recycling, theory of planned behavior, habit, behavior change, container deposit scheme

## Abstract

Objective: Container deposit schemes are often hailed as a useful avenue to increase consumer recycling rates. Yet, there is little research investigating within-person changes in people’s beliefs and behavior following the implementation of these schemes, or tests of the mechanisms by which such change has occurred. Methods: The current study fills this knowledge gap and assessed container recycling behavior and habits as well as the social cognition factors of attitudes, subjective norms, perceived behavioral control, and intentions in a sample of 90 Queenslanders before the implementation of the container deposit scheme and one and three months post-implementation. Results: Analysis of variance indicated more frequent recycling behavior following the implementation of the scheme, as well as stronger habits, intentions, and perceived behavioral control. Conclusions: Such a concomitant change in behavior, beliefs, and habits provides support for behavior change theory, while also flagging potential targets for strategies that can be paired with container deposit schemes to enhance their efficacy and uptake.

## 1. Introduction

A key part of the United Nations strategy for sustainable development is a substantial increase in recycling rates worldwide [1]. In line with these global targets, a key environmental goal that the Australian government set out in the National Waste Policy Action Plan 2019 is to reduce the total waste generated in Australia by 10% per person by 2030, setting a target of recycling or reusing 80% of the country’s waste [2]. Yet, despite clear goals and the accepted need for change, evidence indicates that in the decade preceding 2020, Australia’s recycling rate remained relatively stagnant, and where change has occurred, it has been largely insufficient to meet objectives [3].

One element of recycling identified for improvement by the government in the state of Queensland, Australia, was single-use drink containers (cans, plastic bottles, glass bottles). As of 2018, these containers made up almost half of all litter in the state and were often sent incorrectly to the landfill from household waste collection [4,5]. As part of a concerted effort to improve recycling rates for single-use drink containers, the state government in Queensland introduced the Containers for Change scheme in November 2018—a container deposit scheme where single-use drink containers could be returned to selected locations around the state for an AUD 10c refund. Specifically, the scheme aimed to improve recycling rates by providing a financial incentive for the return of single-use containers, for example, by encouraging households or businesses to correctly sort recycling waste for an additional income stream or allowing community groups to use the removal of litter from public areas as a method of fundraising [5].

Across Australia, these schemes have demonstrated efficacy through strong uptake, reduced litter, and improved recycling rates [5,6,7]. For example, in Queensland, 63.5% of eligible containers were recycled in 2022–2023, with the majority collected via container refund points [8]. However, while there is evidence in favor of container deposit schemes from population-level statistics, there is little research examining within-person longitudinal change or investigations into the factors which underlie increased recycling following the implementation of container deposit schemes. That is, while changes in laws and socio-structural variables are often associated with changes in behavior, it is important to note that changes in people’s behavior are unlikely to occur in isolation but rather because of changes in one’s context being reflected in changes in perceptions toward a given behavior [9]. 

This process of change can be explained by social psychological theory, a prototypical example of which is the theory of planned behavior [10]. The theory of planned behavior posits that the effect of socio-structural variables on behavior is likely mediated by beliefs [10,11]. In the model, such beliefs are summarized as three constructs: attitudes, defined as beliefs that engaging in a behavior will result in positive or negative outcomes and the value attached to those outcomes; subjective norms, defined as beliefs that engaging in a behavior will be considered desirable or undesirable by significant others and the motivation to comply with these desires; and perceived behavioral control, defined as beliefs that engaging in a behavior is under one’s own volitional control and the strength of those beliefs. In the context of pro-environmental and recycling behaviors, the theory of planned behavior has demonstrated significant efficacy in predicting both intentions and behavior [12,13], while qualitative research has flagged the constructs of the theory of planned behavior as potential determinants of willingness to use container deposit schemes in pre-implementation focus groups [14]. 

Broader theoretical research and empirical evidence has shown support for the proposition that socio-structural factors influence behavior through the beliefs encompassed in the theory of planned behavior; meta-analytic structural equation modelling testing the effects of socio-structural variables on numerous behaviors has shown such effects to be at least partially mediated by the beliefs encompassed in the theory of planned behavior [9]. In the current context, adopting the theory of planned behavior presents a potentially valuable avenue of research not only to investigate the effectiveness of the container deposit scheme implemented in Queensland on people’s recycling behavior, but also to test whether the mechanisms for the effectiveness of the scheme reflect behavior change theory. This is because it is unlikely that the implementation of financial incentives alone directly led to an increase in behavior. Instead, it is likely that offering a financial incentive to recycle drinking containers is associated with changes in beliefs about recycling which, in turn, are associated with changes in intentions and behavior. For example, the provision of a financial incentive likely offers a potentially salient additional positive outcome from recycling and thus may lead to more positive attitudes towards recycling behavior. By investigating the concomitant changes in beliefs alongside changes in container recycling behavior, the current research offers a potentially useful avenue for testing behavior change theory while also providing evidence which may directly inform campaigns aiming to increase the efficacy of container deposit schemes. This may be of particular note considering that the launch or expansion of container deposit schemes is currently under consideration in several other locations and populations [3].

Beyond the effect of the container deposit scheme on recycling beliefs and intentions, it is likely that if the container deposit scheme is successful in changing behavior, the altered context will also present an opportunity for the development of new or changed recycling habits [15,16]. That is, as recycling behavior becomes more frequent, particularly if it is undertaken in similar contexts and is viewed as positive, then recycling is likely to become increasingly habitual. By extension, the development of recycling habits likely encourages sustained recycling in future, as supported by the reciprocal relations between habit and behavior over time [17,18]. Such a hypothesis presents an important element of assessing the success of behavior change programs, given that the effects of belief-based behavior change strategies over long periods of time are often called into question [19,20]. In contrast, habits, once developed, are theorized to be deep-seated and highly resistant to change [21,22]. Thus, in the current context, the concomitant changes of habit and behavior following the implementation of the container deposit scheme provides both support for recent habit theory and a potential indicator of the longevity of behavior change which has occurred in the immediate aftermath of legislative changes.

### The Current Study

The current study aimed to use the implementation of a container deposit scheme in Queensland, Australia, as an opportunistic natural experiment, using a longitudinal design to assess the concomitant changes in beliefs, intentions, habits, and behavior following the statewide implementation of a financial incentive for recycling single-use drinking containers. That is, while there is population-level evidence for the awareness and effectiveness of the container deposit scheme in Queensland [5], little is known regarding how these programs effect change in behavior, particularly regarding what theory-based constructs change following the implementation of such schemes and may act as mechanisms of change for increased recycling behavior.

In the current study, it was predicted that, following the implementation of the container deposit scheme, participants would report greater intentions and more favorable beliefs toward recycling drinking containers, would report more frequent recycling of drinking containers, and would be more likely to report recycling drinking containers as habitual. The specific hypotheses were as follows:

**H1.** 
*Attitudes towards recycling drinking containers will be increasingly favorable over time with the implementation of the container deposit scheme.*


**H2.** 
*Subjective norms around recycling drinking containers will be more positive over time with the implementation of the container deposit scheme.*


**H3.** 
*Perceived behavioral control of recycling drinking containers will be stronger over time with the implementation of the container deposit scheme.*


**H4.** 
*Intentions towards recycling drinking containers will increase over time with the implementation of the container deposit scheme.*


**H5.** 
*Participants will report increasingly stronger container recycling habits over time with the implementation of the container deposit scheme.*


**H6.** 
*Container recycling behavior will increase over time with the implementation of the container deposit scheme.*


## 2. Method

### 2.1. Participants and Procedure

The sample consisted of Queensland residents recruited through advertisements on social media. Baseline data were collected from January to October 2018, before the implementation of a container deposit scheme in November 2018. At the baseline measurement point, participants provided data on their demographic information and their beliefs, behavior, and habits around recycling single-use drinking containers after every use via an online survey hosted on the Qualtrics platform. Participants were then recontacted via email in December 2018 to complete the same measures in the period following the implementation of the container deposit scheme and again in February 2019 for follow-up data. At the baseline time-point, 199 participants completed the survey measures. However, 109 did not respond to email requests for follow-up data, leaving a final sample of 90 (see Table 1 for full demographic data). Participants who did not complete the follow-up measures did not significantly differ from the final sample in terms of study variables at baseline (*F*(6, 191) = 1.88, *p* = 0.083, η^2^_p_ = 0.086), although those who did not complete follow-up measures were more likely to be younger (*t*(194) = 6.47, *p* < 0.001) and male (χ^2^(1) = 7.67, *p* 0.006) than the final sample. All procedures were approved by the Griffith University Human Research Ethics Committee.

### 2.2. Measures

All self-report measures are available in full in Table 2.

**Behavior**. Participants recycling single-use drinking containers at each time-point were assessed as the mean of two items (e.g., “How often did you recycle every single-use drinking container you used during the past four weeks?”). Each item was scored on a 7-point scale (e.g., [1] Never to [7] Every Time). The measure had good reliability at each time-point (Time 1: α = 0.98, Time 2: α = 0.98, Time 3: α = 0.98).

**Attitude**. Attitude towards recycling single-use drink containers was assessed as the mean of four items with the common stem “For me to recycle every single-use drinking container I use in the next four weeks would be…”, each scored on a 7-point bipolar scale ([1] Bad to [7] Good). The measure had good reliability at each time-point (Time 1: α = 0.94, Time 2: α = 0.80, Time 3: α = 0.99).

**Subjective Norms**. Subjective norms about recycling single-use drink containers were assessed as the mean of four items assessing a combination of injunctive and descriptive norms (e.g., “Those people who are important to me would want me to recycle every single-use drinking container I use”), each scored on a 7-point Likert scale anchored from [1] Strongly Disagree to [7] Strongly Agree. The measure had good reliability at each time-point (Time 1: α = 0.88, Time 2: α = 0.86, Time 3: α = 0.90).

**Perceived Behavioral Control**. Perceived behavioral control towards recycling single-use drink containers was assessed as the mean of four items (e.g., “It would be easy for me to recycle every single-use drinking container”), each scored on a 7-point Likert scale anchored [1] Strongly disagree to [7] Strongly agree. The measure had good reliability at each time-point (Time 1 α = 0.82, Time 2 α = 0.85, Time 3 α = 0.86).

**Intention**. Intention to recycle each drinking container they used was assessed as the mean of four items (e.g., “I intend to recycle every single-use drinking container I use”), each scored on a 7-point Likert scale anchored from [1] Strongly Disagree to [7] Strongly Agree. The measure had good reliability at each time-point (Time 1: α = 0.94, Time 2: α = 0.89, Time 3: α = 0.96).

**Habits**. Participants’ habits around recycling single-use drinking containers were assessed using the four-item self-reported behavioral automaticity index (e.g., “Recycling every single-use drinking container I use is something I do automatically”) [23,24]. Items were scored on a 7-point Likert scale anchored from [1] Strongly Disagree to [7] Strongly Agree. The measure had good reliability at each time-point (Time 1: α = 0.94, Time 2: α = 0.97, Time 3: α = 0.96).

### 2.3. Data Analysis

Data were fitted to a repeated measures multivariate analysis-of-variance model with three time-points. Following a significant multivariate effect of time, we assessed univariate models for changes over time in attitude, subjective norms, perceived behavioral control, intention, habits, and behavior. Finally, where univariate models indicated a significant change over time, we applied linear polynomial contrasts to assess our hypotheses of continuing changes in each model construct from pre-implementation of the container deposit scheme to post-implementation and three months post-implementation. Power analysis for the three-wave MANOVA indicated a minimum sample of 54, assuming a medium effect size (η^2^_p_ = 0.06) with a required power of 0.80 and an α of 0.05, indicating that the study was adequately powered for this analysis.

## 3. Results

Means and standard deviations for each variable are presented in Table 3. A multivariate analysis of variance indicated a significant effect of time on recycling beliefs and behavior (*F*(6, 84) = 1.95, *p* = 0.041, η^2^_p_ = 0.231). Specifically, in univariate models, there were significant changes in recycling behavior (*F*(2, 178) = 9.87, *p* < 0.001, η^2^_p_ = 0.100), perceived behavioral control (*F*(2, 178) = 2.84, *p* = 0.002, η^2^_p_ = 0.069), intention (*F*(2, 178) = 6.61 *p* = 0.002, η^2^_p_ = 0.069), and habits (*F*(2, 178) = 4.51, *p* = 0.012, η^2^_p_ = 0.048). However, no significant change was observed for attitude (*F*(2, 178) = 0.66, *p* = 0.520, η^2^_p_ = 0.007) or subjective norms (*F*(2, 178) = 1.26, *p* = 0.286, η^2^_p_ = 0.014). Significant effects were investigated through linear polynomial contrasts to assess continued change over time, which indicated a significant positive direction change in mean values over time for behavior (*F*(1, 89) = 19.21, *p* < 0.001, η^2^_p_ = 0.177), perceived behavioral control (*F*(1, 89) = 10.76, *p* = 0.001, η^2^_p_ = 0.108), intention (*F*(1, 89) = 8.16, *p* = 0.005, η^2^_p_ = 0.084), and habits (*F*(1, 89) = 7.33, *p* = 0.008, η^2^_p_ = 0.076).

## 4. Discussion

The current research aimed to investigate changes in recycling beliefs, behaviors, and habits in the immediate aftermath of a statewide implementation of a container deposit scheme in Queensland, Australia, thus flagging potential mechanisms by which this scheme may be influencing household recycling behavior. The results indicated that following the implementation of the container deposit scheme, participants viewed recycling as increasingly under their control, had stronger intentions and habits around recycling each drinking container they used, and reported more frequent recycling of drinking containers. 

As expected, the implementation of the container deposit scheme was associated with a higher reported intention to recycle and a higher self-reported frequency of recycling drinking containers, which is in line with population-level data collected by the Queensland government and environmental groups following the implementation of the scheme [25,26]. Such an effect is likely expected, given the increased incentives available for recycling. However, from a theoretical standpoint, one may also expect that changes in recycling behavior occurred as the result of concomitant changes in beliefs about recycling behavior that stemmed from the implementation of the container deposit scheme [9,10,11]. For example, one may expect that by providing an increased financial incentive to recycle, participants would view recycling as more favorable and perceive greater approval from others of the behavior which, in turn, would lead to increased positive attitudes and subjective norms, respectively. Yet, in the current study, we observed no significant change in either attitude or subjective norms. It is important to consider that for attitude, the lack of a significant effect may be explained by the very positive attitudes towards recycling held by participants at baseline, leaving little room for more favorable attitudes to develop. Regarding the minimal impact on subjective norms, it is possible that despite qualitative research linking container deposit scheme use to normative beliefs [14], the provision of financial incentives for recycling has little effect on changing people’s normative beliefs, given that financial benefits to the self may not directly affect individuals’ perceptions of what others might expect of them. In light of evidence regarding the potential importance of norms in eliciting pro-environmental behaviors, alternative strategies may be needed to target these beliefs [27,28].

In contrast, we observed a significant change in perceived behavioral control, as participants reported higher self-efficacy beliefs towards recycling after the implementation of the container deposit scheme. While it is not possible from the current data to understand exactly which factors led to an increase in perceived behavioral control, one plausible explanation may stem from the campaigns launched alongside the container deposit scheme to encourage business participation [29], for example, by informing businesses about the possibility of placing additional recycling bins for single-use containers in shops, hospitality venues, and offices and depositing containers as an additional source of revenue. Should such campaigns have been successful, it may be that the perceived ease of recycling increased by way of a larger number of recycling bins and more readily available access to them in the general environment following the implementation of the container deposit scheme. However, without detailed qualitative data, such an explanation is entirely speculative, and additional research into self-efficacy beliefs is needed to probe changes in perceived behavioral control.

Alongside a significant change in recycling behavior, participants also reported viewing their recycling of drinking containers as increasingly automatic over time, a key hallmark of the habit construct [30]. This is in line with previous research on legal or social changes prompting changes in habitual behavior over time [31]. In the current context, it could be speculated that while the initial change in recycling behavior following the implementation of the container deposit scheme is likely due to changes in beliefs, an increased frequency of recycling behavior, often carried out in similar contexts, provides additional opportunities for habit development [32]. Further, once developed, the increasing frequency of recycling and the perception of recycling as automatic are likely mutually reinforcing [17,18], a notion supported by the continuing change over time observed in both behavior and habits. Beyond supporting a central tenant of habit theory, the significant change in habits over time also provides a speculative indication for the long-term efficacy of the container deposit scheme, as the formation of recycling habits is likely to be self-sustaining [22] and thus promote recycling behavior in the future. Additional research is needed to confirm whether this is the case.

### Strengths, Limitations, and Future Directions

The current study has several important strengths, including a novel test of the effect of wholesale legislative changes in recycling beliefs and behavior. These data, paired with population-level statistics [5,25,26], provide additional evidence in favor of the uptake of the container deposit scheme in Queensland, Australia. Further, by employing a theory-driven approach to assessing concomitant changes in recycling beliefs and behavior, the current research adds to data on the effectiveness of the scheme by highlighting potential mechanisms of change. This, in turn, provides evidence for effective targets in future messaging to boost the continued uptake of container deposit schemes. From a policy perspective, these findings may be of particular importance, especially given the initial success of the program in Queensland during the data collection period yet the observed stagnation of single-use container deposit rates in subsequent years, which has resulted in a shortfall of the stated goal of an 85% recycling rate [5,8]. The knowledge gained from this initial insight into the changes in beliefs and habits following the implementation of the container scheme provides potential targets for behavior change strategies to bolster the use of the container scheme in Queensland and other jurisdictions with similar programs, for example, through programs fostering positive normative beliefs around using container deposit schemes or encouraging beliefs around the ease of use of container return centers.

However, despite the potential value of the findings from this research, this study is inherently not without limitations, which should be considered when interpreting the current results. First, although the current study presented an opportunistic natural experiment, it is not possible to recruit a matched control group, as changes were implemented on a population-wide level. Thus, definitive conclusions on whether the source of the observed changes was due to the implementation of the container deposit scheme cannot be confirmed from the current findings. These concerns may be partially allayed by comparison with other jurisdictions or behaviors. For example, around the time of data collection, Western Australia was not implementing similar-scale changes to recycling policies and observed little change in per capita recycling rates at a population level [33], and general recycling tonnage did not increase during the 2018–2019 period in Queensland [4]. However, as direct comparison data were not collected, this is speculative and issues of causality can only be addressed through future research (e.g., by comparing behaviors across jurisdictions during differential policy changes). Second, recycling behavior in the current research was assessed using a brief self-report scale. While there is evidence for the validity of similar self-report scales for other health and recycling behaviors [34,35], it is nonetheless a concern that responses may have been not entirely accurate due to self-presentation or recall biases, leading to the under- or overreporting of recycling behaviors. Thus, results from self-reported measures may not reflect real-world recycling behaviors, and the current results should be interpreted with caution. Lastly, the opt-in sample combined with the higher-than-expected attrition may be noteworthy for the external validity concerns of the present research. That is, it is plausible that the final sample included in this study may not be representative of the wider population, and thus, the results in the current research may not replicate on a larger scale. While concerns in this regard are at least partially addressed by the lack of significant differences in baseline beliefs between those who completed the follow-up survey and those who did not, it is important to consider that the potential for selection bias in the results remains. Future research is, thus, recommended, with larger representative samples, which would not only improve the external validity of the current findings but also allow for more complex tests of mediational effects within the theory of planned behavior to assess the mechanisms of action for changes in recycling behaviors. 

## 5. Conclusions

The current study investigated the changes in drink-container recycling beliefs, behaviors, and habits following the introduction of a container deposit scheme in Queensland, Australia. As expected, recycling behaviors, intentions, habits, and perceived behavioral control all increased following the implementation of the scheme, although no change was observed in attitudes or subjective norms. These findings provide overall support for the container deposit scheme, while also showing preliminary evidence for habit and social cognition theories through the expected concomitant changes in beliefs, behaviors, and habits. From a practical perspective, such findings may be of value for informing campaigns paired with the implementation or expansion of container deposit schemes. That is, by understanding the mechanisms by which these strategies are effective in altering behavior, it is possible to design campaigns to amplify such effects in future instances in which similar schemes are launched or expanded. However, considering the modest sample size and use of self-report measures, future research should seek to confirm and expand on the current findings using more-intensive research designs.

## Figures and Tables

**Table 1 behavsci-14-00112-t001:** Sample demographics.

Demographic	Statistic
Mean Age (SD)	41.53 (13.76)
Gender	
Male	77
Female	13
Relationship Status	
Never Married	24
Married or De Facto	56
Separated	8
Widowed	1
Prefer Not to Say	1
Employment	
Full Time	43
Student	8
Part Time	26
Unemployed	12
Prefer Not to Say	1
Education	
High School	5
TAFE Certificate/Diploma	20
Undergraduate Degree	40
Postgraduate Degree	25
Ethnicity	
Caucasian	86
Other	4
Family Taxable Income	
Nil–AUD 18,200	6
AUD 18,201–AUD 37,000	9
AUD 37,001–AUD 80,000	22
AUD 80,000–AUD 180,000	37
AUD 180,000+	13
Prefer Not to Say	3

**Table 2 behavsci-14-00112-t002:** Survey items.

Item	Response
Behavior	
To what extent did you recycle every single-use drinking container you used during the past four weeks?	[1] Never to [7] Every Time
How often did you recycle every single-use drinking container you used during the past four weeks?	[1] Never to [7] Very Frequently
Attitude	
For me to recycle every single-use drinking container I use in the next four weeks would be…	[1] Bad to [7] Good [1] Useless to [7] Useful [1] Worthless to [7] Valuable [1] Harmful to [7] Beneficial
Subjective Norms	
Those people who are important to me would want me to recycle every single-use drinking container I use Most people who are important to me would approve of me recycling every single-use drinking container I use Most people who are important to me think I should recycle every single-use drinking container I use Those people who are important to me do recycle every single-use drinking container they use	[1] Strongly Disagree to [7] Strongly Agree
Perceived Behavioral Control	
It is mostly up to me whether I recycle every single-use drinking container It would be easy for me to recycle every single-use drinking container I have complete control over whether I recycle every single-use drinking container I am confident that I could recycle every single-use drinking container	[1] Strongly Disagree to [7] Strongly Agree
Intention	
I intend to recycle every single-use drinking container I use It is likely that I will recycle every single-use drinking container I use I expect that I will recycle every single-use drinking container I use I am willing to recycle every single-use drinking container I use	[1] Strongly Disagree to [7] Strongly Agree
Habits	
Recycling every single-use drinking container I use is something… …I do automatically …I do without having to consciously remember …I start doing before I realize I’m doing it …I do without thinking	[1] Strongly Disagree to [7] Strongly Agree

**Table 3 behavsci-14-00112-t003:** Descriptive statistics at each time-point.

	T1	T2	T3
Behavior			
*M*	5.87	6.09	6.42
*SD*	1.46	1.35	1.12
95% CIs	5.56–6.17	5.81–6.38	6.19–6.66
Attitude			
*M*	6.74	6.79	6.85
*SD*	0.79	0.84	0.52
95% CIs	6.58–6.91	6.62–6.97	6.74–6.96
Subjective Norms			
*M*	5.45	5.63	5.60
*SD*	1.28	1.14	1.24
95% CIs	5.18–5.72	5.39–5.86	5.34–5.86
Perceived Behavioral Control			
*M*	6.00	6.18	6.35
*SD*	1.29	1.15	1.05
95% CIs	5.73–6.27	5.93–6.42	6.13–6.57
Intention			
*M*	6.27	6.56	6.61
*SD*	1.31	0.85	0.97
95% CIs	6.00–6.55	6.38–6.74	6.41–6.81
Habits			
*M*	5.66	5.91	6.08
*SD*	1.69	1.55	1.46
95% CIs	5.31–6.02	5.59–6.24	5.77–6.38

Note: for comparison, simple pairwise comparisons are available in Appendix A.

## Data Availability

Study data and analysis outputs are available on the Open Science Framework at https://osf.io/qj6sz.

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
