# Peer review of "Can the Cans: Determinants of Container Deposit Behavior before and after Introduction of a Container Refund Scheme"

_behavsci, 2024, doi:10.3390/bs14020112_

Round 1

Reviewer 1 Report

Comments and Suggestions for Authors

Dear Authors

Thank you for submitting your paper to this outlet. I read your paper and gave my concern down here:

1. The originality of the paper can be improved. Particularly, the strengths of the paper can be improved.

2. The research problem is not evidently clear.

3. The methodology section needs further improvement. Particularly, the reliability and validity issues were not described.

4. As the data were cross-sectional, these data needs to undergo CMB issues. 

5. Please describe the strengths of the paper.

Wish them all the best.

Author Response

Reviewer 1

Reviewer’s Comment: The originality of the paper can be improved. Particularly, the strengths of the paper can be improved.

Authors’ Reply: We thank the reviewer for their time in reviewing the manuscript. We have replied to each of the reviewer comments regarding the requested expansion upon the originality and contribution of the paper. Changes in the manuscript are marked in red text.

Reviewer’s Comment: The research problem is not evidently clear.

Authors’ Reply: We have expanded upon this in the “The Current Study” section of the manuscript on Page 3:

“The current study aimed to use the implementation of a container deposit scheme in Queensland, Australia as an opportunistic natural experiment, using a longitudinal design to assess the concomitant change in beliefs, intentions, habits, and behavior following the statewide implementation of a financial incentive for recycling single use drinking containers. That is, while there is population level evidence for the effectiveness of the container deposit scheme in Queensland, little is known regarding how these programs effect change in behavior, particularly regarding what theory-based constructs change following the implementation of such schemes and may act as mechanisms of change for increased recycling behavior.”

Reviewer’s Comment: The methodology section needs further improvement. Particularly, the reliability and validity issues were not described.

Authors’ Reply: We agree on the value of this data and have thus included additional information in this regard to the manuscript. For example, reliability data in the form of Cronbach’s alpha is now included in the measures section for each scale.

Reviewer’s Comment: As the data were cross-sectional, these data needs to undergo CMB issues.

Authors’ Reply: The data for this manuscript was not cross-sectional, but consisted of longitudinal data over three time points. We have now made this clear in the “Current Study” section on page 3.

Reviewer’s Comment: Please describe the strengths of the paper.

Authors’ Reply: We have expanded upon the future direction section of this paper, which is now named “Strengths, Limitations, and Future Directions”. This section now includes a paragraph summarizing the strengths of this paper on page 8.

“The current study has several important strengths, including a novel test of the effect of wholesale legislative changes on recycling beliefs and behavior. This data, paired with population level statistics [22,23], provides additional evidence in favor of the uptake of the container deposit scheme in Queensland, Australia. Further, by employing a theory driven approach to assessing concomitant change in recycling beliefs and behavior, the current research adds to data on the effectiveness of the scheme by highlighting potential mechanisms of change. This, in turn, provides evidence for effective targets in future messaging to boost continued uptake of container deposit schemes.”

Reviewer 2 Report

Comments and Suggestions for Authors

The study tackles an important real-world issue, namely, how to promote increasing recycling rates for containers, and tests the impact of a statewide container deposit scheme. This is a relevant issue with substantial policy and practical implications.

To this purpose, the study employs a longitudinal pre-post design spanning before and after implementation of the scheme, allowing for stronger causal inferences about the impact of the scheme. Multiple constructs from prominent behavior change theory (theory of planned behavior) are assessed, enabling a test of theoretical mechanisms of change.

However, in its current version the study presents some problems that need to be addressed for it to be publishable. I list them below.

1. No clear hypotheses is stated to guide analyses and interpretations. Without a clear theoretical benchmark, it is difficult to develop a clear interpretation of the results and to reconstruct the underlying rationale.

2. Self-report measures of behavior are notoriously prone to biases. More objective measures would strengthen confidence in findings. The study would be considerably strengthened if self- reported measures would be accompanied by some form of indirect but objectively observable measurement of behavioral change, e.g., local recycling statistics.

4. Is the multivariate ANOVA sufficiently powered given the small sample size? More generally, sample size is a clear limitation of this study. Univariate tests for each outcome variable would be advisable.

5. Effect sizes for significant effects are mostly small-medium, and again this is not very encouraging for such a small sample study.

6. The discussion is lengthy and repetitive in parts (e.g. habit sections). More concision is needed.

7. How the paper’s findings fit with past similar research on deposit schemes? This is not discussed in the level of detail that would be advisable. Additionally, the study’s novelty compared to prior studies is unclear.

Overall, the study makes a modest contribution in demonstrating behavior change mechanisms in a real-world recycling context. It is true that, with the above listed limitations, it provides some initial evidence for possible efficacy of a deposit scheme via habit development. However, strong claims about the scheme's actual impact do not seem warranted in view of the study’s limitations. Replications in larger samples seem really necessary to this purpose.

The authors should therefore provide a more compelling case about why the study, with its current limitations, makes a significant enough contribution to the literature to warrant publication.

Author Response

Reviewer 2

Reviewer’s Comment: The study tackles an important real-world issue, namely, how to promote increasing recycling rates for containers, and tests the impact of a statewide container deposit scheme. This is a relevant issue with substantial policy and practical implications. To this purpose, the study employs a longitudinal pre-post design spanning before and after implementation of the scheme, allowing for stronger causal inferences about the impact of the scheme. Multiple constructs from prominent behavior change theory (theory of planned behavior) are assessed, enabling a test of theoretical mechanisms of change. However, in its current version the study presents some problems that need to be addressed for it to be publishable. I list them below.

Authors’ Reply: We thank the reviewer for their comments and their time in considering this manuscript. We have replied to each of the suggestions below, with changes in the main manuscript marked in red text.

Reviewer’s Comment: 1. No clear hypotheses is stated to guide analyses and interpretations. Without a clear theoretical benchmark, it is difficult to develop a clear interpretation of the results and to reconstruct the underlying rationale.

Authors’ Reply: Thank you for this comment. We agree and have now included the theory-based study hypotheses underpinning the study. This is now reflected in the Introduction and Current Study sections, with changes marked in red text. Primarily, we have reworked the hypotheses section into more explicit numbered hypotheses to more accurately reflect the underlying rationale and aid in interpretation of the study design and results, on Page 3:

“The current study aimed to use the implementation of a container deposit scheme in Queensland, Australia as an opportunistic natural experiment, using a longitudinal design to assess the concomitant change in beliefs, intentions, habits, and behavior following the statewide implementation of a financial incentive for recycling single use drinking containers. That is, while there is population level evidence for the effectiveness of the container deposit scheme in Queensland, little is known about how these programs effect change in behavior, particularly regarding what theory-based constructs change following the implementation of such schemes and may act as mechanisms of change for increased recycling behavior.

In the current study, it was predicted that following the implementation of the container deposit scheme, participants would report greater intentions and more favorable beliefs toward recycling drinking containers, would report more frequent recycling of drinking containers, and would be more likely to report recycling drinking containers as habitual. Specific hypotheses are as follows:

H1: Attitude towards recycling drinking containers will be increasingly favorable over time with the implementation of the container deposit scheme.

H2: Subjective norm around recycling drinking containers will be more positive over time with the implementation of the container deposit scheme.

H3: Perceived behavioral control of recycling drinking containers will be stronger over time with the implementation of the container deposit scheme.

H4: Intention towards recycling drinking containers will increase over time with the implementation of the container deposit scheme.

H5: Participants will report increasingly stronger container recycling habits over time with the implementation of the container deposit scheme.

H6: Container recycling behavior will increase over time with the implementation of the container deposit scheme.”

Reviewer’s Comment: 2. Self-report measures of behavior are notoriously prone to biases. More objective measures would strengthen confidence in findings. The study would be considerably strengthened if self- reported measures would be accompanied by some form of indirect but objectively observable measurement of behavioral change, e.g., local recycling statistics.

Authors’ Reply: In the current study we assessed recycling behavior for individual participants and thus we do not believe local recycling statistics would not be an appropriate measure of any given individuals’ behavior. While similar self-reported behavior measures have been used in the past for other behaviors with acceptable correlations with observational measures, we acknowledge that this is none-the-less a concern for the current data. We have expanded on the potential implications for this in the discussion section on Page 8:

“Second, recycling behavior in the current research was assessed using a brief self-report scale. While there is evidence for the validity of similar self-report scales for other health and recycling behaviors [31,32], it is none-the-less a concern that responses may have been not entirely accurate due to self-presentation or recall biases.”

 Further, we now note in the discussion section the alignment of our findings with larger scale evidence from other research and the Queensland government for the effectiveness of the scheme:

“As expected, the implementation of the container deposit scheme was associated with a higher reported intention to recycle and a higher self-reported frequency of re-cycling drinking containers, in line with population level data collected by the Queensland Government and environmental groups following the implementation of the scheme [19,20].”

Reviewer’s Comment: 4. Is the multivariate ANOVA sufficiently powered given the small sample size? More generally, sample size is a clear limitation of this study. Univariate tests for each outcome variable would be advisable.

Authors’ Reply: The power analysis presented in the original submission was in reference to the MANOVA analysis presented. However, we recognize this may not have been entirely clear, so have now expanded on our power analysis and moved it to a new section 2.3 Data Analysis on Page 6.

“Power analysis for the three-wave MANOVA indicated a minimum sample of 54, assuming a medium effect size (η2p= .06) with a required power of .80 and an α of .05.

In the manuscripts current form, we present a MANOVA test for overall effects across time, followed by univariate tests of each effect, as is often recommended by experts in quantitative methodology. As the univariate tests requested by the reviewer are already presented in the results section as follow-up analyses to the MANOVA and the MANOVA is adequately powered, we do not believe there is sufficient cause to remove the omnibus MANOVA test.

Reviewer’s Comment: 5. Effect sizes for significant effects are mostly small-medium, and again this is not very encouraging for such a small sample study.

Authors’ Reply: Similar to the last point, we have now expanded our presentation of the power analysis to make specific reference to the MANOVA employed to test effects in the data. In our a-priori power analysis we used an expected effect size of η2p= .06. We believe in light of the results this was a reasonable hypothesized effect, as the observed effect for the MANOVA was well above this hypothesized range (η2p= .231), while where change was observed in follow-up univariate analyses, effects were of a similar size (η2p= .048 to .100).

Reviewer’s Comment: The discussion is lengthy and repetitive in parts (e.g. habit sections). More concision is needed.

Authors’ Reply: We have now revised parts of the discussion section to avoid repetition and with a more in depth focus on the implications of the current data. Changes in this area are marked throughout in red text.

Reviewer’s Comment: 7. How the paper’s findings fit with past similar research on deposit schemes? This is not discussed in the level of detail that would be advisable. Additionally, the study’s novelty compared to prior studies is unclear.

Authors’ Reply: We have expanded the Introduction and Discussion sections of the manuscript to include an expanded literature review, including additional citations on past research in similar research areas, such as the research predicting new recycling schemes and research following changes in beliefs and behaviors following legislative changes. For example on Page 2:

“In the context of pro-environmental and recycling behaviors, the theory of planned behavior has demonstrated significant efficacy in predicting both intentions and behavior [9,10], while qualitative research has flagged the theory of planned behavior constructs as potential determinants of willingness to use container deposit schemes in pre-implementation focus groups [11].”

Reviewer’s Comment: Overall, the study makes a modest contribution in demonstrating behavior change mechanisms in a real-world recycling context. It is true that, with the above listed limitations, it provides some initial evidence for possible efficacy of a deposit scheme via habit development. However, strong claims about the scheme's actual impact do not seem warranted in view of the study’s limitations. Replications in larger samples seem really necessary to this purpose. The authors should therefore provide a more compelling case about why the study, with its current limitations, makes a significant enough contribution to the literature to warrant publication.

Authors’ Reply: We agree with the reviewer of ensuring study conclusions are matched with data findings. In light of this suggestion and similar notes by other reviewers, we have now expanded the limitations and future directions section on Page 8, and have edited our conclusions throughout the Discussion section to ensure consistency with the presented data.

Reviewer 3 Report

Comments and Suggestions for Authors

This paper reports data from a survey conducted before and after the implementation of a deposit refund scheme. The topic is interesting, and the paper contributes to the literature as there seem to be few studies with similar methods. The paper is clearly written and easy to follow. I have a few comments that I would like the authors to address, in particular about the method and results section of the manuscript.

Introduction: the introduction is well-written and goes straight to the point.

Methods:

1)      Could the authors provide more information on their sample? They mention it was recruited on social media, but was “recycling” mentioned as the theme of the survey? If so, probably only people interested by the topic registered to participate in the survey. Given the average level at T1 on behavior, it is possible that it is only people who already recycle at least part of their waste. My point is that, besides the small size of the sample which is already mentioned as a limitation, I would like to see more information in the method (and also addressed in the discussion) about the sample composition, how much is it representative of the general population? This is important in terms of the external validity of the results.

2)      One item supposed to measure subjective norm does not match the definition of the concept (“those people who are important to me do recycle every single-use drinking container they use”, it reflects what others do, not what they expect the participant to be doing). Was a reliability analysis conducted (e.g., cronbach alpha) before creating the scores? This item should probably be left out of the SN score.

Results:

3)      The descriptive statistics at each time point suggest that the changes appear at different time points (notably, for behavior, it seems to be increasing between T2 and T3, while for intention, the increase seems to be from T1 to T2). I would like to have more information on those differences, and if they are meaningful, to see some discussions about it.
In relation to this, I wondered why polynomial contrasts were chosen, was it hypothesized? In its current form, the contrast analysis does not really add much to the main analysis given that the pattern is not described.

Discussion:

4)      I appreciate that the authors recognize the fact that in the absence of control group, the changes cannot be attributed for sure to the new scheme. However, in order to improve the reach of the paper, I wondered if it could be possible to add some data about the recycling trend in Queensland for the study time period, and compare it to other Australian states recycling trend at the same time. This data might be available a posteriori, and it would allow to understand if the change is due to this policy happening in that states, or to some more global trend (e.g., increase in environmental awareness) that might happen in other states as well.

Author Response

Reviewer 3

Reviewer’s Comment: This paper reports data from a survey conducted before and after the implementation of a deposit refund scheme. The topic is interesting, and the paper contributes to the literature as there seem to be few studies with similar methods. The paper is clearly written and easy to follow. I have a few comments that I would like the authors to address, in particular about the method and results section of the manuscript.

Authors’ Reply: We thank the reviewer for their feedback and time in assessing this manuscript. We have replied to each of their comments below with corresponding changes in the manuscript marked in red text.

Reviewer’s Comment: Introduction: the introduction is well-written and goes straight to the point.

Authors’ Reply: We thank the reviewer for their positive feedback.

Reviewer’s Comment: Could the authors provide more information on their sample? They mention it was recruited on social media, but was “recycling” mentioned as the theme of the survey? If so, probably only people interested by the topic registered to participate in the survey. Given the average level at T1 on behavior, it is possible that it is only people who already recycle at least part of their waste. My point is that, besides the small size of the sample which is already mentioned as a limitation, I would like to see more information in the method (and also addressed in the discussion) about the sample composition, how much is it representative of the general population? This is important in terms of the external validity of the results.

Authors’ Reply: We agree and have expanded upon the “2.1 Participants and Procedure” section to provide additional information on the composition of the study sample in Table 1. We have also expanded upon our discussion of the value of collecting a larger, more representative sample in the “Strengths, Limitations, and Future Directions” section on Page 8:

“Lastly, the opt-in sample combined with higher-than-expected attrition may be noteworthy for the external validity concerns of the present research. While concerns in this regard are at least partially addressed by the lack of significant differences on baseline beliefs between those who completed the follow-up survey and those who did not, it is important to consider that the potential for selection bias in the results remains. Future research is thus recommended, with larger representative samples which would not only improve the external validity of the current findings but also allow for more complex tests of mediational effects within the theory of planned behavior to assess mechanisms of action for changes in recycling behaviors.”

Reviewer’s Comment: One item supposed to measure subjective norm does not match the definition of the concept (“those people who are important to me do recycle every single-use drinking container they use”, it reflects what others do, not what they expect the participant to be doing). Was a reliability analysis conducted (e.g., cronbach alpha) before creating the scores? This item should probably be left out of the SN score.

Authors’ Reply: We thank the reviewer for pointing out this issue. We have now included reliability statistics in the form of Cronbach’s alpha for each measure in the 2.2 Measures section on page 4. As the reviewer correctly points out, we used a measure of subjective norms that includes items capturing elements of both descriptive and injunctive norms. While this may not be as common as measuring injunctive norms alone, it is consistent with the recommendations for the reasoned action approach [1]. We now explicitly note this in the measures section for subjective norms on Page 4. Notably, despite including both injunctive and descriptive norm related items, the scale reliability remained acceptable at each time point, with Cronbach’s alpha statistics between .86 and .90.

Reviewer’s Comment: The descriptive statistics at each time point suggest that the changes appear at different time points (notably, for behavior, it seems to be increasing between T2 and T3, while for intention, the increase seems to be from T1 to T2). I would like to have more information on those differences, and if they are meaningful, to see some discussions about it. In relation to this, I wondered why polynomial contrasts were chosen, was it hypothesized? In its current form, the contrast analysis does not really add much to the main analysis given that the pattern is not described.

Authors’ Reply: We thank the reviewer for this comment and agree that additional information on the use of polynomial contrasts is warranted. To address this, we have added a new subsection to the methods of the paper, 2.3 Data Analysis. Specifically, in this case, we hypothesized that the introduction of a container deposit scheme would result in a gradual change towards more favorable beliefs and recycling behaviors over time. Thus, we deemed the fitting of linear contrasts representing a continuing trend in improvement to be the most appropriate analysis, as compared to simple contrast between time points. This is now presented on Page 6:

“Data were fitted to a multi-variate analysis of variance model with three time points. Following from a significant multi-variate effect of time, we assessed univariate models for each change over time for attitude, subjective norm, perceived behavioral control, intention, habit, and behavior. Finally, where univariate models indicated a significant change over time, we applied linear polynomial contrasts to assess our hypotheses of continuing changes in each model construct from pre-implementation of the container deposit scheme, post implementation, and three months post implementation.”

However, we acknowledge that readers may contrasts valuable for a more in-depth investigation in effects, have now included an alternative analysis of effects using simple contrasts as a supplement. Note that in the now included supplement, the general pattern of effects remains consistent, with a modest increase in behavior, perceived behavioral control, intention, and habit over the course of the study.

 Reviewer’s Comment: I appreciate that the authors recognize the fact that in the absence of control group, the changes cannot be attributed for sure to the new scheme. However, in order to improve the reach of the paper, I wondered if it could be possible to add some data about the recycling trend in Queensland for the study time period, and compare it to other Australian states recycling trend at the same time. This data might be available a posteriori, and it would allow to understand if the change is due to this policy happening in that states, or to some more global trend (e.g., increase in environmental awareness) that might happen in other states as well.

Authors’ Reply: We thank the reviewer for this interesting suggestion. Although it is not possible to make true comparisons, we now make note in the discussion section of recycling rates in Western Australia for a comparable time period. Western Australia was selected as New South Wales and Victoria were also implementing a container deposit scheme at the time, while South Australia already had a container deposit scheme, and Tasmania was implementing a broader recycling initiative. Thus, Western Australian data in this context perhaps represents the cleanest comparison point attainable, although as we mention, only speculative and very tentative. This is now presented on Page 8:

“These concerns may be partially allayed by comparison with other jurisdictions, for example Western Australia which was not implementing similar scale changes to re-cycling policies around the time of data collection, and observed little change in per capita recycling rates at a population level [30]. However, as direct comparison data was not collected this is speculative and can only be addressed in future research comparing behavior across jurisdictions during differential policy changes.”

Reviewer 4 Report

Comments and Suggestions for Authors

This manuscript is on a timely and important issue. It is generally well-written and concise. However, two major concerns prevent my recommending it for publication at this time:

1. Prior literature on recycling and its predictors (e.g., the work of P. Wesley Schultz and Robert Cialdini) is not included, and is foundational.

2. The theory of planned behavior posits mediation in a covariance structure model, but that does not appear to be what is tested, and thus while referenced was not directly tested, and should be. This may necessitate a larger sample size.

Author Response

Reviewer 4

Reviewer’s Comment: This manuscript is on a timely and important issue. It is generally well-written and concise. However, two major concerns prevent my recommending it for publication at this time:

Authors’ Reply: We thank the reviewer for their time and effort in reviewing this manuscript. We have endeavoured to reply to each of your comments below, with changes to the manuscript marked in red text.

Reviewer’s Comment: Prior literature on recycling and its predictors (e.g., the work of P. Wesley Schultz and Robert Cialdini) is not included, and is foundational.

Authors’ Reply: We have tried to incorporate other prior literature as suggested from the reviewer, and only where appropriate.

Reviewer’s Comment: The theory of planned behavior posits mediation in a covariance structure model, but that does not appear to be what is tested, and thus while referenced was not directly tested, and should be. This may necessitate a larger sample size.

Authors’ Reply: This is a good point. A mediation model testing the mechanisms of change would undoubtedly be a strong contribution to the literature. However, as the reviewer suggested, this analysis would require a significantly larger sample of which we were not able to access. Given the value of the suggestion, we now note this in the future directions section as an avenue for additional research which may expand upon the presented findings. See page 8:

“Future research is thus recommended, with larger representative samples which would not only improve the external validity of the current findings but also allow for more complex tests of mediational effects within the theory of planned behavior to assess mechanisms of action for changes in recycling behaviors. “

Round 2

Reviewer 1 Report

Comments and Suggestions for Authors

Congratulations.

Author Response

We thank the reviewer for their time in reviewing this manuscript.

Reviewer 2 Report

Comments and Suggestions for Authors

I appreciate the effort put by the authors in revising the paper according to my comments and suggestions. However, for the paper to be published, it seems to me that some further work is needed, as the authors did not fully address the concerns raised in the previous report in full.

First of all, given the relatively limited level of originality of the methodology and of novelty of the results with respect to the existing literature, the main value added of the study is probably the focus on the Queensland case study. In this regard, the treatment of the case study in the paper is too sketchy. We need to know in some more detail what has been the process that led to the adoption of the scheme, its policy rationale, and an assessment of the scheme’s actual impact, possibly based upon specific data, and more generally a sketch of broader trends in waste management and recycling in Queensland. Such kind of information helps a lot to make sense of the paper’s results and in assessing their scientific and policy interest.

The methodological limitations of the paper should be assessed more clearly. It is acknowledged that since the scheme was applied at the population level it has been impossible to recruit an appropriate control group, but as a consequence of this one cannot conclude that there is a causal relationship between the scheme and the observed changes. Moreover, it should be better stressed the possibility of a potential selection bias resulting from the opt-in sample and high attrition rates. It should also be better highlighted that the use of brief self-report scales is in itself a considerable further potential source of bias. All these concerns should consequently be reflected in the interpretation of the findings and in the assessment of the paper’s scientific contribution.

Finally, from a paper with this focus, one would expect some more specific policy recommendations and actionable insights, to make a significant contribution to the policy debate.

I think it is in the interest of the authors and of the journal to make their analysis more compelling and scientifically relevant. Please consider my remarks as a sympathetic and constructive contribution in this regard.

Author Response

REVIEWER’S COMMENT: I appreciate the effort put by the authors in revising the paper according to my comments and suggestions. However, for the paper to be published, it seems to me that some further work is needed, as the authors did not fully address the concerns raised in the previous report in full.

AUTHORS’ REPLY: We thank the reviewer for their time in reviewing our revised manuscript, and are sorry for any comments the reviewer believed we did not adequately address. We have replied to each of the suggestions below, with changes in the manuscript marked in red text.

REVIEWER’S COMMENT: First of all, given the relatively limited level of originality of the methodology and of novelty of the results with respect to the existing literature, the main value added of the study is probably the focus on the Queensland case study. In this regard, the treatment of the case study in the paper is too sketchy. We need to know in some more detail what has been the process that led to the adoption of the scheme, its policy rationale, and an assessment of the scheme’s actual impact, possibly based upon specific data, and more generally a sketch of broader trends in waste management and recycling in Queensland. Such kind of information helps a lot to make sense of the paper’s results and in assessing their scientific and policy interest.

AUTHORS’ REPLY: We thank the reviewer for this comment, and we have now expanded upon the background of the scheme on Page 1.

“One element of recycling identified for improvement by the Government in the state of Queensland, Australia was single use drink containers (cans, plastic bottles, glass bottles). As of 2018, these containers made up almost half of all litter in the state and were often sent incorrectly to landfill from household waste collection [4,5]. As part of a concerted effort to improve recycling rates for single use drink containers, the State Government in Queensland introduced the Containers for Change scheme in November 2018; a container deposit scheme where single use drink containers  could be returned to selected locations around the state for a AU10c refund. Specifically, the scheme aimed to improve recycling rates by providing a financial incentive for the return of single use containers, for example by encouraging households or businesses to correctly sort recycling waste for an additional income stream or allowing community groups to use the removal of litter from public areas as a method of fundraising [5].”

We are hesitant to make detailed comparisons to overall waste management and recycling data, as our data focused on the recycling of single use drinking containers only. In contrast, Queensland’s publicly accessible recycling data is assessed on an annual basis and household recycling is aggregated beyond single use drinking containers. Thus, we believe any direct comparisons to specific state recycling data would be too speculative to provide meaning. Instead, where possible, we have included additional information from reviews on the container deposit schemes itself, which focus on the rate of return of eligible containers (see page 2 of the revised paper).

“Across Australia, these schemes have demonstrated efficacy through strong uptake, reduced litter, and improved recycling rates [5–7]. For example, in Queensland, 63.5% of eligible containers were recycled in 2022-2023, with the majority collected via container refund points [8].”

REVIEWER’S COMMENT: The methodological limitations of the paper should be assessed more clearly. It is acknowledged that since the scheme was applied at the population level it has been impossible to recruit an appropriate control group, but as a consequence of this one cannot conclude that there is a causal relationship between the scheme and the observed changes. Moreover, it should be better stressed the possibility of a potential selection bias resulting from the opt-in sample and high attrition rates. It should also be better highlighted that the use of brief self-report scales is in itself a considerable further potential source of bias. All these concerns should consequently be reflected in the interpretation of the findings and in the assessment of the paper’s scientific contribution.

AUTHORS’ REPLY: We agree with the reviewer on the value of this information. Each of these points has been addressed in the “Strengths, Limitations, and Future Directions” section. However, as suggested by the reviewer, we have expanded or rephrased each of the points mentioned, with additional text marked in red text. Please see “Strengths, Limitations, and Future Directions” section on page 8 for changes.

REVIEWER’S COMMENT: Finally, from a paper with this focus, one would expect some more specific policy recommendations and actionable insights, to make a significant contribution to the policy debate.

AUTHORS’ REPLY: This is an important comment and we have now expanded upon our discussion of the study findings for policy recommendations and actions. Please see “Strengths, Limitations, and Future Directions” section on page 8.

"From a policy perspective, these findings may be of particular importance especially given the initial success of the program in Queensland during the data collection period yet the observed stagnation of single use container deposit rates in subsequent years, which has resulted in a short fall of the stated goal of an 85% recycling rate [5,8]. The knowledge gained from this initial insight into the changes in beliefs and habit following the implementation of the container scheme, provides potential targets for behavior change strategies to bolster use of the container scheme in Queensland and other jurisdictions with similar programs. For example, through programs fostering positive normative beliefs around using container deposit schemes or encouraging beliefs around the ease of use of container return centers."

REVIEWER’S COMMENT: I think it is in the interest of the authors and of the journal to make their analysis more compelling and scientifically relevant. Please consider my remarks as a sympathetic and constructive contribution in this regard.

AUTHORS’ REPLY: We thank the reviewer again for their time in reviewing this manuscript.

Reviewer 3 Report

Comments and Suggestions for Authors

I thank the authors for their work on the paper. If I am not mistaken, the supplementary analysis, made available on OSF, are not cited in the paper. A mention of those supplementary analysis should be added in the result section. Otherwise, the paper seems to me ready for publication.

Author Response

We thank the reviewer for their time. We have now added mention of the supplementary analyses to the manuscript.

Reviewer 4 Report

Comments and Suggestions for Authors

Much improved, and satisfied my major concerns. 

(Small one: author Schultz, reference 24, is PW, not WP)

Author Response

We thank the reviewer for their time and have corrected the error in reference 24.

Round 3

Reviewer 2 Report

Comments and Suggestions for Authors

I think the authors satisfactorily addressed my remaining concerns. The papr is ready for publication.